



# Constraining the ship contribution to the aerosol of the Central Mediterranean

S. Becagli[1], F. Anello[2], C. Bommarito[2], F. Cassola[3,4], G. Calzolai[5], T. Di Iorio[6], A. di Sarra[6], J.L. Gómez-Amo[6,7], F. Lucarelli[5], M. Marconi[1], D. Meloni[6], F. Monteleone[2], S. Nava[5], G. Pace[6], M. Severi[1], D. M. Sferlazzo[8], R. Traversi[1], and R. Udisti[1]

[1] Department of Chemistry, University of Florence, Sesto Fiorentino, Florence, I-50019, Italy
[2] ENEA, Laboratory for Earth Observations and Analyses, Palermo, I-90141, Italy
[3] Department of Physics & INFN, University of Genoa, Genoa, I-16146, Italy.
[4] ARPAL-Unità Operativa CFMI-PC, Genova, I-16129, Italy.
[5] Department of Physics, University of Florence & INFN-Firenze, Sesto Fiorentino, Florence, I-50019, Italy.
[6] ENEA Laboratory for Earth Observations and Analyses, Roma, I-00123, Italy
[7] Department of Earth Physics and Thermodynamics, University of Valencia, Spain.
[8] ENEA, Laboratory for Earth Observation and Analyses, Lampedusa, I-92010, Italy

**Keywords**: ship aerosol, Central Mediterranean Sea, $PM_{10}$, La-Ce ratio, Vanadium.

## Abstract

$PM_{10}$ aerosol samples were collected during summer 2013 within the framework of the Chemistry and Aerosol Mediterranean Experiment (ChArMEx) at two sites located North (Capo Granitola, 36.6°N, 12.6°E) and South (Lampedusa Island, 35.5°N, 12.6°E), respectively, of the main Mediterranean shipping route in the Sicily Channel.

The $PM_{10}$ samples were collected with 12 hour time resolution at both sites. Selected metals, main anions, cations, and elemental and organic carbon were determined.

The evolution of soluble V and Ni concentrations (typical markers of heavy fuel oil combustion) was related to meteorology and ship traffic intensity in the Sicily Channel, using a high resolution regional model for back trajectories calculation. Elevated concentration of V and Ni were associated with transport from the Sicily Channel and coincidences between trajectories and positions of large ships, both at Capo Granitola and Lampedusa; the vertical structure of the planetary boundary layer also appears to play a role, with high V values associated with strong inversions and stable boundary layer. The V concentration was generally lower at Lampedusa than at Capo Granitola, where it reached a peak value of 40 $ng/m^3$.

Concentrations of rare earth elements, La and Ce in particular, were used to identify possible contributions from refineries, whose emissions are also characterized by elevated V and Ni amounts; refinery emissions are expected to display high La/Ce and La/V ratios, due to the use of La in the fluid catalytic converter systems. In general, low La/Ce and La/V ratios were observed in the PM samples, allowing to unambiguously identify the large role of the ship source in the Sicily Channel.




Based on the sampled aerosols, ratios of the main aerosol species arising from ship
emission with respect to V were estimated with the aim of deriving a lower limit for the
total ship contribution to $PM_{10}$. The estimated minimum ship emission contributions to
$PM_{10}$ was 1.9 $\mu g/m^3$ at Lampedusa, and 2.8 $\mu g/m^3$ at Capo Granitola, corresponding to
11% and 8.2% of $PM_{10}$, respectively.

## 1. Introduction

Ship emissions may significantly affect atmospheric concentrations of several important
pollutants, especially in maritime and coastal areas (e.g. Endresen et al., 2003). Main
emitted compounds are carbon dioxide ($CO_2$), nitrogen oxides ($NO_x$), sulfur dioxide
($SO_2$), carbon monoxide (CO), hydrocarbons, and primary and secondary particles.
Thus, ship emissions impact the greenhouse gas budget (Stern, 2007), acid rain
(through $NO_x$ and $SO_2$ oxidation products; Derwent at al., 2005), human health (CO,
hydrocarbons, particles; Lloyd's Register Engineering Services, 1995; Corbett et al.,
2007) and solar radiation budget through aerosol direct and indirect effects (black
carbon and sulfur containing particles; Devasthale et al., 2006; Lauer et al., 2007;
Coakley and Walsh, 2002).
Heavy oil fuels used by ships contain varying transition metals originating from the fuel.
The aerosol emitted by ship engines is formed at high temperature (>800°C) from V,
Ni, Fe compounds (Sippula et al., 2009). The thermodynamics predict that these
species mainly form oxides, but when the flue gas dew point is reached, sulfuric acid
(which was found to form a liquid layer on the ultra-fine particles) condenses on it
leading to partial dissolution of the ultra-fine seeds, probably increasing the toxicity of
the particles when inhaled.
In spite of the great amount of gas and particulate arising from ship emission, maritime
transport is relatively clean if calculated per kilogram of transported material, and it is
currently increasing with respect to air and road transport (Micco and Pérez, 2001;
Grewal and Haugstetter, 2007). In addition, emissions from other transport sectors are
decreasing due to the implementation of advanced emission reduction technologies,
and the relative impact of shipping emissions is increasing.
Regulations aiming at reducing emissions based on restrictions on the fuel sulfur
content (sulfur emission control areas, SECAs) have been implemented in several
regions. Although the legislation is focussed on sulfur emissions, the overall health and
environmental effects of the emissions depend in a complicated manner on the physical
and chemical properties of the emissions (WHO, 2013). Several studies have been
carried out to determine the detailed chemical composition of shipping emissions
(Agrawal et al., 2008a and b, Moldanová et al., 2009, Murphy et al., 2009, Lyyränen et
al 1999, Cooper, 2003, Sippula et al. 2014); however, in comparison to on-road
vehicles, the ships emissions are still poorly characterized.
A large variety of anthropic sources (refineries, power plants, intense ship traffic), also
associated with a high population density, and natural emissions make the



Mediterranean region one of the most polluted in the world (e.g., Kouvarakis et al.,
2000; Marmer and Langmann, 2005). This multiplicity of Mediterranean sources (some
of which with the same markers of ship aerosol) makes difficult the quantification of
ship contribution to the total aerosol amount (e.g., Becagli et al., 2012).
The contribution of ships and harbour emissions to local air quality, with specific focus
on atmospheric aerosol, has been investigated using models (Trozzi et al., 1995;
Gariazzo et al., 2007; Eyring et al., 2005; Marmer et al., 2009), experimental analyses
at high temporal resolution (Ault et al., 2010; Contini et al., 2011; Jonsson et al., 2011;
Diesch et al., 2013; Donateo et al., 2014), receptor models based on identification of
chemical tracers associated with ship emissions (Viana et al., 2009; Pandolfi et al.,
2011; Cesari et al., 2014), and integrated approaches with receptor and chemical
transport models (Bove et al., 2014). Few studies exist in open sea (Becagli et al.,
2012; Schembari et al., 2014; Bove et al., 2016).
In this context, studies performed at Mediterranean sites, where it is possible to
distinguish ship emission from other sources of heavy fuel oil combustion, are important
to investigate the current impact of the ship emissions on primary and secondary
aerosols. In a previous study (Becagli et al., 2012) we used measurements of $PM_{10}$ and
relative chemical composition carried out at Lampedusa, in the central Mediterranean,
to investigate the role of ship emissions. Vanadium and Nickel were used as tracers of
heavy fuel combustion together with trajectory analyses to assess the role of ship
traffic. The ship source, however, could not be unequivocally separated from possible
influences from refineries and power plants, which use similar fuels. In summer 2013
we addressed the same topic by implementing a specific strategy to target the aerosols
due to ship emissions. $PM_{10}$ samples were collected in parallel at Lampedusa (LMP) and
at Capo Granitola (CGR), i.e., respectively South and North of the main shipping route
through the Mediterranean, with the aim of isolating the ship source. Figure 1 shows
the map of the measurement stations in the central Mediterranean; Capo Granitola is
about 230 km North of Lampedusa. The analysis is complemented with measurements
of Rare Earth Elements (REEs), trajectories from a high resolution regional model, and
actual observations of ship traffic. The combination of these approaches allows
unambiguously identifying and providing constraints for the ship contribution to $PM_{10}$ in
the central Mediterranean.
The $PM_{10}$ samples were collected in summer 2013 as a contribution to the Chemistry
and Aerosol Mediterranean Experiment (ChArMEx; http://charmex.lsce.ispl.fr).
Lampedusa is one of the supersites of the ChArMEx experiment; a list of the
instruments deployed during the special observing period 1a of ChArMEx, and of the
measurement strategy, meteorological conditions, and main observations is given by
Mallet et al. (2016).



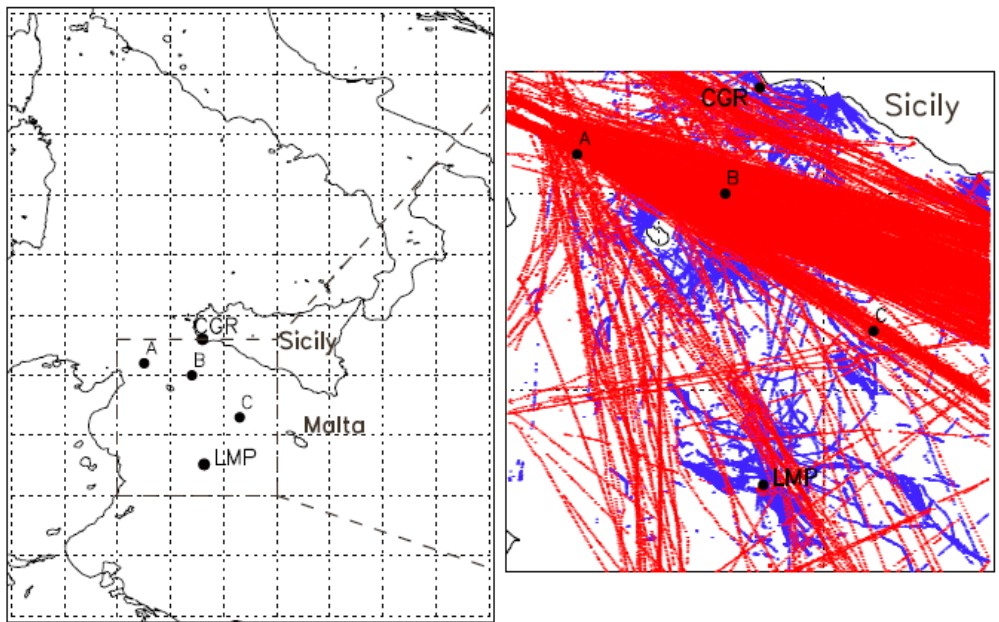

Figure 1. Map of the study area with the sites of Lampedusa (LMP) and Capo Granitola (CGR) (left panel). A, B and C indicate the three sites selected to study the stability of the boundary layer in the Sicily Channel (see section 3.2.2). The ship routes in the study area during the first 10 days of June 2013 are displayed in the right panel. Red and blue dots show the routes of merchant and fishing vessels, respectively.

## 2. Measurements and methods

### 2.1. Aerosol sampling and chemical analyses

$PM_{10}$ was sampled at two sites: at the Station for Climate Observations, maintained by ENEA (the Italian Agency for New Technologies, Energy, and Sustainable Economic Development) on the island of Lampedusa (35.5°N, 12.6°E), and at the Italian CNR (National Research Council) Research Centre at Capo Granitola (36.6°N, 12.6° E ). Lampedusa is a small island in the Central Mediterranean sea, more than 100 km far from the nearest Tunisian coast. At the Station for Climate Observations, which is located on a 45 m a.s.l. plateau on the North-Eastern coast of Lampedusa, continuous observations of greenhouse gases concentration (Artuso et al., 2007, 2009), aerosol properties (di Sarra et al., 2011, 2015; Becagli et al., 2013; Marconi et al., 2014; Calzolai et al., 2015), total ozone, ultraviolet irradiance (Meloni et al., 2005), solar and infrared radiation (di Sarra et al., 2011; Meloni et al., 2012, 2015), and other climatic parameters are carried out.





$PM_{10}$ is routinely sampled on a daily basis at LMP (Becagli et al., 2013; Marconi et al.,
2014; Calzolai et al., 2015). For the intensive ChArMEx campaign, samples were
collected from 1 June to 3 August at 12-hour resolution by using a low volume dual
channel sequential sampler (HYDRA FAI Instruments) equipped with two $PM_{10}$ sampling
heads operating in accord with UNI EN12341.
The two channels operated in parallel and were loaded with different types of filters:
the first one with 47 mm diameter, 2 μm nominal porosity Teflon filters, for ion
chromatographic analysis of soluble ions, atomic emission spectroscopy for soluble
metals, and proton-induced X-ray emission (PIXE) for the total (soluble+insoluble)
elemental composition; the second one with 47 mm pre-fired, 2 μm nominal porosity,
quartz filters for elemental (EC) and organic carbon (OC) determinations.
The sampling site at CGR is located at Torretta Granitola, a Research Center of the
Italian National Research Council, in South-Western Sicily (12 km from Mazara del
Vallo). The sampler was installed on the roof of one of the research centre buildings at
about 20 m a.s.l., directly on the coastline, facing the strait of Sicily.
At CGR $PM_{10}$ samples were collected at 12 hour resolution with a TECORA Skypost
sequential sampler on 47 mm pre-fired, 2 μm nominal porosity, quartz filters allowing
the determination of ions, metals, EC and OC on different fractions of the filter. Due to
technical problems, some diurnal samplings were lost at CGR.
The $PM_{10}$ mass was determined by weighting the filters before and after sampling with
an analytical balance in controlled conditions of temperature (20±1 °C) and relative
humidity (50±5 %). The estimated error on $PM_{10}$ mass is around 1% at 30 μg/m$^3$ in the
applied sampling conditions.
A quarter of each Teflon filter from LMP and a 1.5x1 cm punch of the quartz filter from
CGR were analysed by Ion Chromatography (IC) in the analytical conditions described
in Marconi et al. (2014). The estimated uncertainty for IC measurements is 5% for all
the considered ions.
Blank values were negligible with respect to the concentration in the samples for Teflon
filters. Blank values for quartz filters were negligible for most of the analyzed species,
and when not negligible, anyway lower than 25$^{th}$ percentile, they were subtracted from
the measured concentrations.
Another quarter of the Teflon filter from LMP, and another 1.5x1 cm punch of the
quartz filter from CGR were extracted in ultrasonic bath for 15 min with MilliQ water
acidified at pH 1.5–2 with ultrapure $HNO_3$ obtained by sub-boiling distillation. This
extract was used for the determination of the metals soluble part by means of an
Inductively Coupled Plasma Atomic Emission Spectrometer (ICP-AES, Varian 720-ES)
equipped with an ultrasonic nebulizer (U5000 AT+, Cetac Technologies Inc.). The
chosen value of pH is the lowest found in rainwater (Li and Aneja, 1992) and leads to
the determination of the metals fraction available to biological organisms and, for some
metals (e.g. V and Ni), related to the anthropic source (Becagli et al., 2012).
The remaining half Teflon filter from Lampedusa was analysed by proton induced X ray
emission (PIXE) technique (Lucarelli et al., 2011). PIXE analysis is a non-destructive



method for metals. Thus, after the PIXE analysis, this part of the Teflon filter and
another punch of the quartz filter from CGR were used for the determination of metals
by ICP-AES through the solubilisation procedure according with the method reported in
the EU EN14902 (2005) rule, by using concentrated sub-boiling distilled $HNO_3$ and 30%
ultrapure $H_2O_2$ in a microwave oven at 220°C for 25 min (P = 55 bar). Although this
solubilisation procedure is not able to completely dissolve the silicate species, is able to
recover at least 70% of the same elements measured by PIXE also for elements having
dominant crustal source (unpublished data) due to the low crustal aerosol load in these
sampling period (e.g., Mailler et al., 2016).
The OC and EC measurements were carried out on a 1.5x1 cm punch of the quartz
filters from Lampedusa and Capo Granitola by means of a Sunset thermo-optical
transmittance analyser, following the NIOSH protocol (Wu et al, 2016).

## 2.3. Atmospheric model and trajectory calculations

Numerical simulations with a non-hydrostatic mesoscale atmospheric model were used
to characterize the meteorological conditions in the Sicily Channel during the campaign
and to support the interpretation of the experimental results. The Weather Research
and Forecasting (WRF) model (Skamarock et al., 2008) outputs, provided by the
Department of Physics of the University of Genoa, Italy, were used, covering the entire
Mediterranean with a grid spacing of 10 km and hourly temporal resolution. Initial and
boundary conditions to drive WRF simulations are obtained from the Global Forecast
System operational global model (Environmental Modeling Center, 2003) outputs
(0.5x0.5 square degree). Some recent applications of the modelling chain are described
in Mentaschi et al. (2015) and Cassola et al. (2016), where full details on the model
configuration can also be found.
In particular, the WRF 3-D hourly meteorological fields were used to perform a
backward trajectory analysis with the NOAA HYbrid Single-Particle Lagrangian
Integrated Trajectory Model (HYSPLIT; Stein et al. 2015), aimed at assessing the origin
of air masses impacting the monitoring sites and at supporting the source attribution
suggested by the analysis of specific markers (see, in particular, Section 3.2.2). The use
of a high-resolution regional atmospheric model for trajectory calculations allows a
better representation of boundary layer properties and mesoscale phenomena such as
land/sea breezes, which can have a relevant impact especially in complex topography
coastal sites like CGR.
Specifically, 48-h long back trajectories were computed at each site from a reference
height of 10 m above ground level, starting every six hours for the whole period of the
campaign, from 10[th] June to 31[st] July 2013.

## 2.4. Ships/marine traffic






The position and the main characteristics of the ships travelling in the central
Mediterranean were derived from the MarineTraffic database
(http://www.marinetraffic.com/), which provides the position of the ships with a high
temporal resolution (about 3-5 minutes) by means of the Automatic Identification
System (AIS).
Three classes of ships defined by the AIS classification were considered: all the ships,
the merchant (i.e. cargo and tanker), and the fishing vessels. The merchant and fishing
vessels are the most frequent ships in the Sicily Channel and the former are expected to
produce the highest impact on the V concentration due to their higher emissions
(http://ec.europa.eu/environment/archives/air/pdf/chapter2_ship_emissions.pdf).
**3. Results**
**3.1. $PM_{10}$ chemical composition at the two sites**
The sea salt aerosol (SSA) component of $PM_{10}$ was estimated as the sum of the sea salt
(ss) fractions of $Na^+$, $Mg^{2+}$, $Ca^{2+}$, $K^+$, sulfate and chloride. Details on the calculation of
sea salt $Na^+$ and $Ca^{2+}$, and non-sea salt (nss) fractions are reported in Marconi et al.
(2014). The sea salt fractions of $Mg^{2+}$, $Ca^{2+}$, $K^+$, and sulphate were calculated from sea
salt $Na^+$ ($ssNa^+$) by using the ratio of each component to $Na^+$ in bulk sea water:
$Mg^{2+}/Na^+ = 0.129$, $Ca^{2+}/Na^+ = 0.038$, $K^+/Na^+ = 0.036$, $SO_4^{2-}/Na^+ = 0.253$ (Bowen,
1979). Chloride undergoes depletion processes during aging of sea spray, mainly due
to exchange reactions with anthropic $H_2SO_4$ and $HNO_3$, leading to re-emission of HCl in
the atmosphere. Thus, for chloride we use the measured chloride concentration instead
of the one calculated from $ssNa^+$. Thus,
$$SSA = 1.46 *[ssNa^+] + [Cl^-]$$
The crustal component is calculated from Al, which represents 8.2% of the upper
continental crust, UCC (Henderson and Henderson 2009). A previous study using an
extensive data set at Lampedusa showed that the crustal content determined from the
total Al was in very good agreement with calculations made from the sum of the metal
oxides (Marconi et al., 2014). However, in this study we use measurements of the
soluble Al concentration obtained by ICP-AES on the solution obtained with $H_2O_2$ and
$HNO_3$ in microwave oven, instead of the total Al content. Therefore, in this work we
underestimate the crustal contribution by about 30% (unpublished results). However, it
must be emphasized that the crustal aerosol contribution was very low throughout the
measurement campaign.
Figure 2 shows the time series of the main $PM_{10}$ components at LMP and CGR. It must
be noticed that an intense Mistral event occurred from 22$^{nd}$ June to 1$^{st}$ July. Mistral





events are characterized by strong winds from the north-westerly sector, and often by
subsiding air masses originating from the free troposphere. Thus, elevated values of
SSA and low concentrations of other compounds are generally found during Mistral.
Average concentrations of $PM_{10}$ and of the different aerosol components for the whole
measurement campaign and for the non-Mistral conditions are reported in Table 1. The
averages were calculated over a homogeneous dataset, i.e., only for the 12-hour
intervals with observations at both sites.

The largest $PM_{10}$ values were linked to elevated SSA during the Mistral event at both
sites. $PM_{10}$ is about two times larger at Capo Granitola than at Lampedusa. The $PM_{10}$
measured during the campaign at Lampedusa was significantly smaller than its long-
term average (31.5 µg/m$^3$; Marconi et al., 2014).  No Saharan dust transport events
occurred at low altitude in this period (e.g., Mailler et al., 2015), and the crustal aerosol
contribution remained very low and almost constant at both sites (average < 1 µg/m$^3$
at LMP and around 3 µg/m$^3$ at CGR).

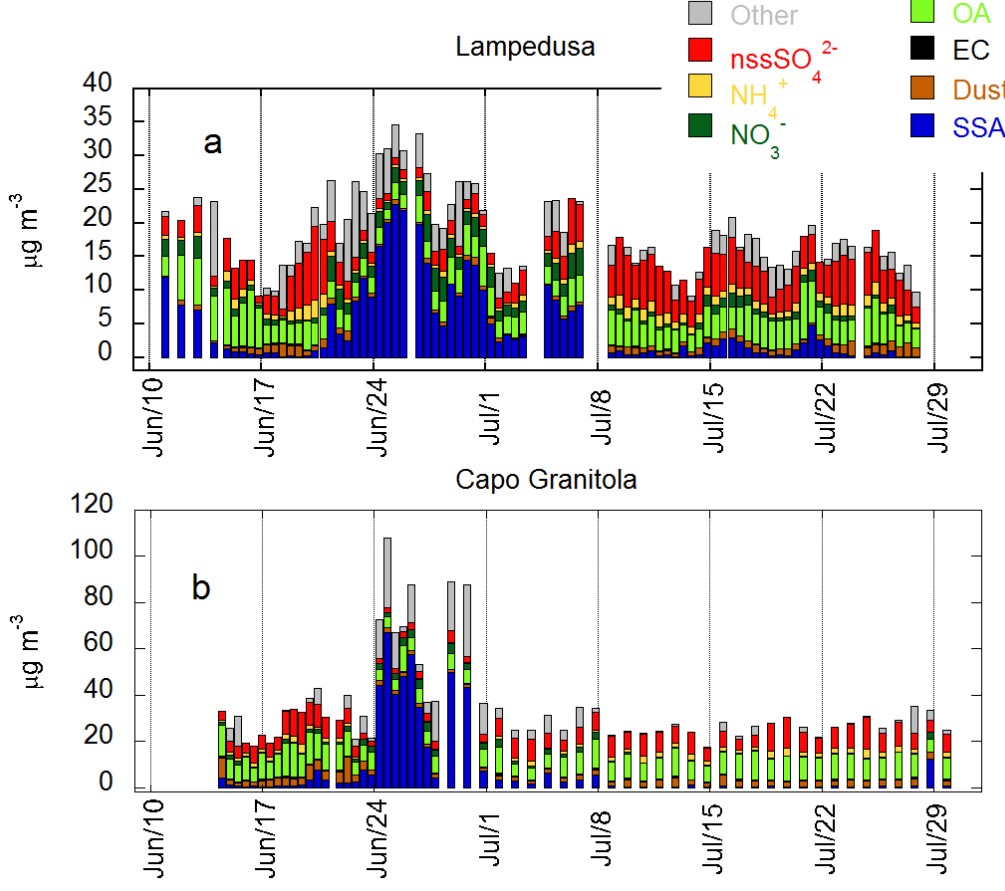




299 Figure 2. Time series of the main aerosol components at LMP (plot a) and CGR (plot b).
300 Note the different vertical scales of the graphs. For calculation of Organic Aerosol (OA)
301 see text.

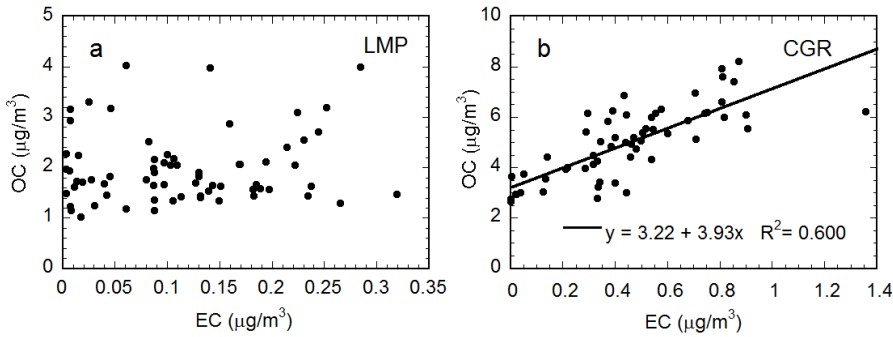

323 Figure 3. Scatter plot of OC vs. EC at LMP (plot a) and CGR (plot b). Note the different
324 scales of the graphs.
328 SSA accounted for about 26% and 24% of $PM_{10}$ at LMP and CGR, respectively. During
329 the periods non influenced by the Mistral the SSA contribution was about 14% at LMP
330 and 8% at CGR.  Non-sea salt $SO_4^{2-}$ was the most abundant among the secondary
331 inorganic species.
332 Organic aerosol was the most abundant component at CGR, where its mean
333 concentration was > 9 $\mu g/m^3$ and represented 35% of $PM_{10}$ in the days not
334 characterized by Mistral. EC was about 4 times higher at CGR than at LMP.
335 Elemental carbon (EC) and organic carbon (OC) displayed a quite different behavior at
336 the two sites. Figure 3 shows the behavior of OC versus EC at LMP and CGR. OC is
337 correlated with EC ($R^2$=0.60; n =59) at CGR, suggesting a strong influence from carbon
338 species primary sources, which are characterized by the simultaneous emission of EC
339 and OC. At LMP, on the contrary, OC was not correlated with EC, indicating a strong
340 impact of OC secondary and/or natural sources.






Thus, we used a conversion factor of 1.8 (typical for urban background sites, Turpin
and Lin, 2001) at CGR, and a conversion factor of 2.1 (typical for remote sites
characterized by high impact of secondary sources, Turpin and Lin, 2001) at LMP to
estimate the total organic aerosol (OA) amount from the measured values of OC. Once
estimated OA with this method, the sum of the various species accounted to more than
85% of the measured mass at both sites. The unreconstructed mass could be due to an
underestimation of OA from OC, or to the presence of bound water not removed by the
desiccation procedure at 50% relative humidity (Tsyro, 2005; Canepari et al., 2013).

Figure 4 shows the combined evolution of nssSO$_4^{2-}$, OA, and V at CGR between 14 and
21 June, based on the 12-hour resolution data. As discussed above, OA was mainly due
to inland primary sources, while V was expected to be mainly associated with ship
emissions (Becagli et al., 2012). Non-sea salt SO$_4^{2-}$ sources are expected to be present
both on land and on sea. It is interesting to note the diurnal cycle at CGR in the period
14-18 June (Figure 4).  The daily cycle is very likely related to the sea breeze regime
which is expected to play a significant role at CGR, and a negligible role at LMP, due to
the island small size.
In the days with dominant sea breeze regime, air masses are advected from the sea
during daytime, and displayed low values of OA and elevated values of V. In nighttime,
when land air masses were driven to the coast, OA was larger, and V lower than in
daytime.















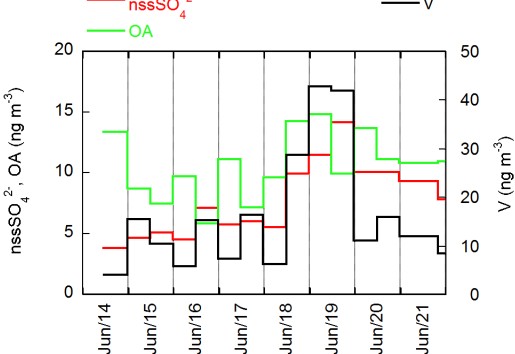

Figure 4. Time series of nssSO$_4^{2-}$, organic aerosol (OA), and V$_{sol}$ at CGR.


### 3.1.1. Ship emission markers: V and Ni


Several studies focussed on the identification of specific tracers of shipping emissions
(Viana et al., 2008; Becagli et al., 2012, Isakson et al., 2001, Hellebust et al., 2010).



Vanadium and Nickel are generally considered the best markers for this source because, after sulfur, they are the main impurities in heavy fuel oil (Agrawal et al., 2008a and b). The soluble fraction of these metals is even more representative for the ship source (Becagli et al., 2012).

Following Becagli et al. (2012), we used measurements of V and Ni soluble fractions ($V_{sol}$ and $Ni_{sol}$, respectively). In the data set here considered the $V_{sol}$ and $Ni_{sol}$ ratio with respect to Al were always more than 10 times larger than for UCC, as expected for cases dominated by heavy oil combustions sources (ships, refineries, power plants, stainless steel production plants).

Figure 5 shows the time series of the V soluble fraction at LMP and CGR. Table 2 reports slope, correlation coefficient and number of samples of the linear correlation between $V_{sol}$ and $Ni_{sol}$.

$V_{sol}$ and $Ni_{sol}$ are highly correlated, suggesting a common source. The obtained slope of the regression line (2.8-2.9, that increases to 3.0 for samples with $V_{sol}$ >6 ng/m$^3$) is typical for heavy fuel oil combustion sources (Mazzei et al., 2008; Agrawal et al., 2008a and b, Viana et al. 2009; Pandolfi et al., 2011). The same value was found at Lampedusa by Becagli et al. (2012), considering data from 2004 to 2008. The behaviour of V, Ni, and their ratio are then representative of heavy fuel oil combustion. It is however difficult to distinguish V and Ni originating from power plants, refineries, or ship engines. Moreover, several refineries are present in Sicily (Siracusa, Gela, Milazzo) and in Sardinia (Cagliari) which may potentially influence the sampling sites.

A combination of methods is thus used in this study to unequivocally identify the ship source. The analysis is based on: additional chemical tracers, like the Rare Earth Elements, whose behaviour is specific for the refinery and the ship sources; high resolution back-trajectories, based on data from the high resolution regional model; information on the vertical mixing in the atmospheric boundary layer; coincidences between the high resolution back-trajectories and the position of different types of ships in the Sicily Channel.

### 3.1.2. Rare Earth elements

As discussed above, anthropic V and Ni originate from heavy oil combustion, and may be considered markers of the ship source only when other sources can be excluded. Few studies propose the use of lanthanoid elements (La to Lu) to distinguish refinery from ship emissions (Moreno et al. 2008). In particular, the ratio between the La and Ce concentrations (La/Ce ratio, hereafter LCR) has been used to identify specific sources. Shipping emissions are characterised by values of LCR between 0.6 and 0.8, similarly to the earth crust. Conversely, elevated values of LCR (from 1 to 5) are associated with emissions from refinery zeolitic fluid catalytic converter plants (Moreno et al., 2008).


LCR at LMP and CGR was generally around the value expected for UCC (Handerson and
Handerson, 2009), also in events with high $V_{sol}$ concentration (fig. 5).
At Capo Granitola LCR was >1 in 10% of the samples, and >2 in 3% of samples. At
Lampedusa 24% of samples displayed LCR>1, and 8% LCR>2. Although during these
days with LCR>1 the concentration of $V_{sol}$ is usually low, a contribution of aerosol from
refinery cannot be excluded.
The behaviour of V, La, and Ce is shown in a 3-component plot in figure 6. La and Ce
were scaled in order to have the typical UCC composition in the central part of the plot.
By comparison, the typical UCC composition and that of uncontaminated and La-
contaminated (Refinery) Asian dust collected at Mauna Loa, Hawai'i by Olmez and
Gordon (Olmez and Gordon, 1985) are also displayed in figure 6.
The data points from LMP and CGR are grouped in a region with elevated values of V,
and marked different amounts of La and Ce with respect to V, differently from UCC and
refineries.

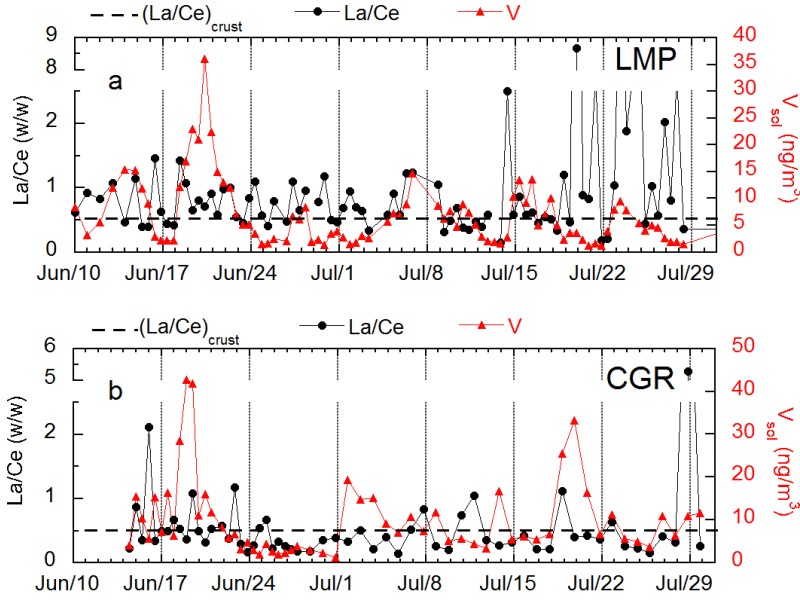

Figure 5. Time series of LCR and V at a), Lampedusa, and b), Capo Granitola. The
horizontal black lines in each plot represent the LCR in the upper continental crust (0.5
w/w).





Figure 6. Three-component Ce-La-V plots for LMP (left) and CGR (right). The UCC composition is marked with a red triangle in the centre of the plots. The blue dot represents the composition of refinery-contaminated Asian dust (Olmez and Gordon, 1985). The black encircled area represents the ship emission composition.

The behaviour of the different chemical tracers support the conclusion that the V due to ship emissions is largely dominant in the $PM_{10}$ measured at LMP and CGR during the measurement campaign. Thus, cases with elevated V can be used to identify cases with a large contribution from the ship source.

**3.2. Trajectories and ship traffic**

**3.2.1 Origin of air masses during the campaign**

All the trajectories arriving at LMP and CGR, calculated with the HYSPLIT model driven by the WRF meteorological fields (see Section 2.3), are shown in an aggregated way in Figure 7, where the trajectory frequency at each point of the computing grid is shown for the whole period (upper panels) and for the June 10th – June 30th interval (lower panels). While at LMP the trajectory frequency pattern is quite elongated in the NW-SE



direction, at CGR trajectories are distributed over a wider range of directions, despite a
general prevalence of northerly sectors. The predominance of air masses coming from
the northwest is particularly evident in June (lower panels), when areas with trajectory
frequencies exceeding 10% are found farther to the north, up to the Gulf of Lion.
During the first part of the campaign (June 2013), indeed, the synoptic situation was
characterized by a "dipolar" sea level pressure anomaly pattern, with positive anomalies
in the western Mediterranean and negative ones in the eastern part of the basin
(Denjean et al., 2016). This situation induced stronger and more frequent than usual
north-westerly winds (i.e. Mistral episodes, see Section 3.1) over the Sardinia and Sicily
Channels.

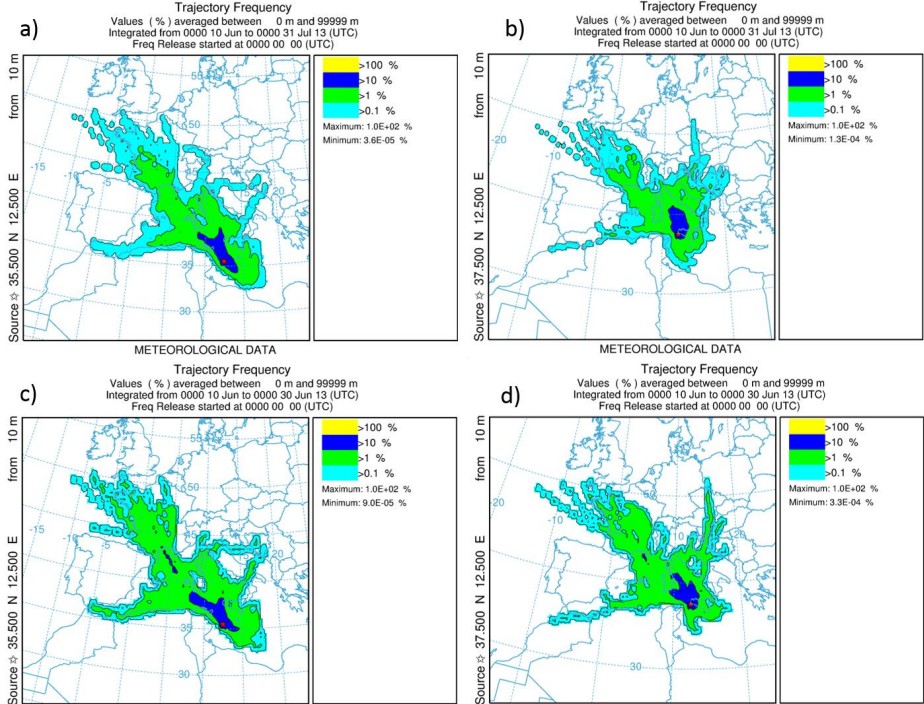

Figure 7. Trajectory frequency computed at each grid cell with starting points at LMP
(panels a,c) and CGR (b,d). Upper panels show values averaged over the whole period
of the campaign (10$^{th}$ June – 31$^{st}$ July 2013), while lower panels are relative to the June
10$^{th}$ –June 30$^{th}$ interval.
**3.2.2 Ship traffic**



To further investigate the mechanisms determining the V enhancement at the two sites
we investigated the relationships among the amount of V, the back-trajectory pattern,
the effective number of ships influencing the air mass, and the stability of the boundary
layer in the ship source region (i.e., the Sicily Channel).
All back-trajectories arriving at LMP and CGR were considered and all trajectory-ship
coincidences occurring within the last 36 hours before sampling were taken into
account
It was assumed that the ship plume influenced the sampled air mass if:
- the trajectory passed at less than 15 km from the position of the ship
- the altitude of the air mass was lower than 500 m.
The total number of ships fulfilling these criteria was associated to each trajectory. The
analysis was based on the available 1-hour time resolution meteorological fields (a ship
influencing a trajectory was counted once every hour).
To further explore the impact of different types of ships, the analysis was carried out
considering the following three ship categories: all the ships, the merchant (i.e. cargo
and tanker), and the fishing vessels.
The atmospheric stability is also expected to play a large role in modulating the V
amounts (Becagli et al. 2012). A temperature inversion, TI, index, was calculated based
on the 3D atmospheric fields from the WRF model at three sites in the Sicily Channel.
The temperature inversions have been used as a proxy to identify the periods
characterized by a stable boundary layer. The three sites, A (37.2°N, 11.5°E), B
(37.0°N, 12.4°E), and C (36.3°N, 13.3°E), were selected in the regions of most
frequent ship passage and crossing with the trajectories from LMP and CGR. The TI
index was calculated as the difference between the temperature at the altitude of the
maximum T, and the surface T. A positive TI indicates an inversion, and the TI value
provides an indication of the intensity of the inversion. Only positive values are
considered in this analysis.
Figure 8 summarizes the results of this analysis. It shows the times series of the
number of the ships influencing the trajectories arriving at LMP and CGR, respectively,
and the corresponding measured values of V. Results are shown for the three classes of
ships. The TI intensity is also shown.
In general, there is a rather good correspondence between the measured values of V
and the number of ships encountered along the associated air mass trajectory at CGR.
The correspondence is somewhat less evident at LMP. As discussed above, the V
concentration is generally higher at CGR than at LMP. Part of this difference may be
ascribed to the shorter distance between CGR and the main shipping route crossing the
Sicily Channel with respect to Lampedusa, and the consequent larger number of
encountered ships.



The analysis of the event with elevated V concentration at both sites, between 18th and
21st June, provides further information to understand the link between the ship traffic
and measured V concentration. This is the only event observed almost simultaneously
at both stations. Maxima of V occurred between 19th and 20th June at CGR (about 42
ng/m$^3$), and on 21st June at LMP (36.1 ng/m$^3$). It is worth noting that similar
concentrations were measured at CGR also around 18th-19th July, in conjunction with an
increase in the number of merchant vessels.
On the other hand, the episode of 19th-21st June is the largest occurring at LMP, both
for duration and V concentration. Especially at the beginning of the event, large values
of V do not correspond with an increase of the number of ships along the air mass
trajectories.
A possible explanation for this behavior is provided by the temporal evolution of TI in
the Sicily Channel. The temperature inversion started to develop on 14 June, and
gradually increased in intensity until 22 June; the TI persistence and progressive
increase in intensity provided suitable conditions for the trapping of the ship plumes in
the boundary layer, with a consequent build-up of the ship aerosol and V concentration.
This process appears particularly efficient at CGR between 21 and 25 June.
A similar combined dependency on number of ships and TI appears also at LMP around
7 July. It is interesting to note that V seems to depend more directly on the number of
merchant ships (see, e.g., the lack of V peaks on 17 June, 12 and 29 July at LMP, when
the number of fishing vessels was high and the number of merchant ships was low)
than on the total or the fishing ships.
Thus, the trajectory analysis carried out in combination with the available information
on the ship tracks confirms that the ship emissions are the main responsible for the
moderate and elevated values of V measured at LMP and CGR during the campaign.
This analysis also clearly suggests that the boundary layer structure plays a very
important role in determining the impact produced by the emissions. This simplified
approach confirms the importance to carefully characterize the emission scenario and
the meteorological conditions in studies on the impact of ships emissions on the air
quality.



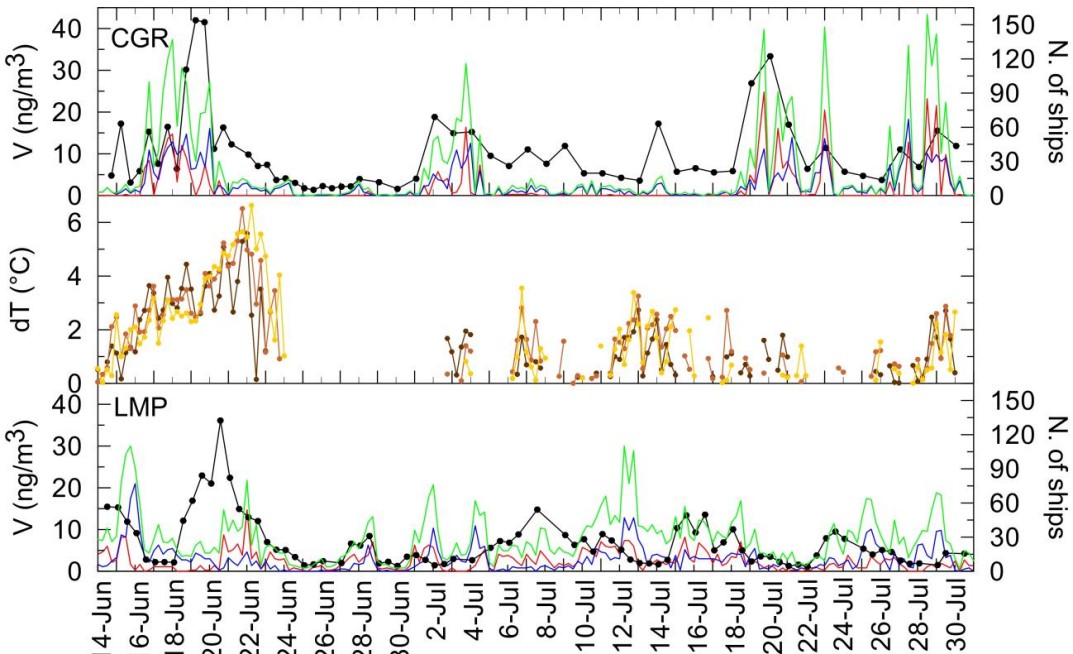

Figure 8. Time series of Vanadium concentration (black line with dots) and number of ships affecting the air masses sampled at CGR (upper panel) and LMP (lower panel). Green, red and blue lines indicate respectively the total number of ships, the number of merchant (i.e. cargo and tanker), and of fishing vessels. The time evolution of the temperature inversion index (dT in the figure) at three different locations in the Sicily Channels is shown in the middle panel; brown, red, and yellow curves show the behavior at sites A, B, and C (see text).

## 3.3. Sulfate, nitrate, and organic carbon from ships

$SO_2$ is one of the main species emitted in the ship plume in the gas phase (Agrawal et al., 2008a, b). $SO_2$ is produced through oxidation of the S contained as impurity in heavy fuel oil, and is an aerosol precursor.

A previous study performed at Lampedusa over 5 years (Becagli et al., 2012) showed that the behavior of non-sea salt sulfate is not directly correlated with V and Ni because several other $SO_4^{2-}$ sources (anthropic, marine biogenic, crustal, volcanic) contribute to the non-sea salt sulfate in the Central Mediterranean Sea.

The same study suggests a lower limit of about 200 for the $nssSO_4^{2-}/V$ ratio for particles originating from heavy oil combustion at Lampedusa.





Figure 9 shows $nssSO_4^{2-}/V$ versus V at LMP and CGR. At both sites $nssSO_4^{2-}/V$ decreases
for increasing V and reaches a lower limit of about 200 at elevated values of V (> 15
$ng/m^3$). We assume that the ship emission is the dominant source of the sampled
particles for these cases with elevated V. This implies that in these cases virtually all
sulfate originated from the ship source, and the observed lower limit for $nssSO_4^{2-}/V$ can
be considered the lower limit for the sulfate to V ratio in the ship plume.
We use a value of 200 in this work as a rough estimate of the sulfate to V ratio, based
on the values obtained in the previous study and confirmed by the data set used in this
study for two sites and reported in figure 9a and b. The $nssSO_4^{2-}/V$ limit values appears
similar at LMP and CGR confirming the reliability of such values for the central
Mediterranean Sea.



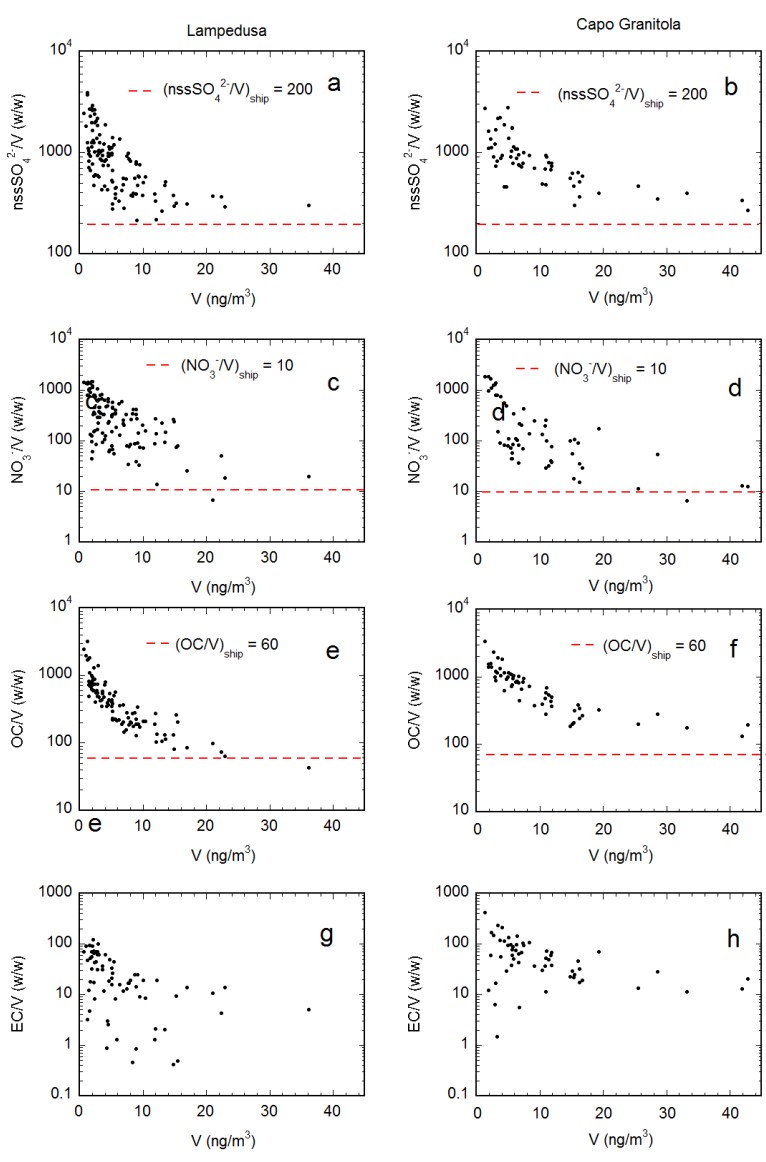

Figure 9. Scatter plots of nssSO$_4^{2-}$/V (a and b), NO$_3^-$/V (c and d), OC/V (e and f) and
EC/V (g and h) vs. V concentration at LMP (plots on the left) and CGR (plots on the
right) sites. The dashed lines in the plot represent lower limits for the characteristic
ratio in the ship plume.





NOx are among the main compounds emitted in gas phase acting as aerosol precursors.
The photochemistry of $NO_x$ leading to $NO_3^-$ formation in the particulate phase is
complex, especially in summer due to the presence of high amounts of OH radical (see
e.g., Chen et al., 2005), and the NOx contribution to the particulate phase is not easy
to be quantified.
Here we try to use the same approach used for sulfate for the determination of a lower
limit for the $NO_3^-$/V ratio in the ship plume.
Figure 8 (plot c and d) shows the $NO_3^-$/V ratio versus V at the two sites. Similarly to
sulfates, the $NO_3^-$/V ratio tends to a lower limit value (around 10 for V higher than 15
$ng/m^3$) at both sites. The NOx concentration is about two times larger than that of $SO_2$
in the ship plume close to the source (Agrawal et al 2008b) and lifetime of NOx is
extremely low (1.8 hour during day and 6.5 hour during night, Chen et al., 2005).
However, the $NO_3^-$/V limit ratio values is low compared to the limit ratio for $SO_4^{2-}$. It has
to be considered that $NO_3^-$ takes part in other photochemical atmospheric reactions that
lead to its removal. Besides, the presence of $HNO_3$ in gas phase not neutralized by $NH_3$
or by sea salt could explain the low $NO_3^-$/$nssSO_4^{2-}$ ratio in the aerosol. Indeed, the $NO_3^-$
concentration measured at LMP and CGR is 4-6 times lower than that of $nssSO_4^{2-}$ (table
1). Low amount of $NO_3^-$ with respect to $SO_4^{2-}$ from ship emissions are found in model
simulations in Southern California (Dabdub, 2008). Indeed, Dabdub (2008) shows that
the contribution to aerosol from ship emissions is 0.05% for $NO_3^-$, and 44% for $SO_4^{2-}$.
Elemental and Organic Carbon are also present in the ship plume (Shah et al., 2004). In
particular, OC constitutes about 15-25% and EC is generally lower than 1% of the PM
sampled at the plume of main ship engine powered by heavy fuel oil (Agrawal et al.,
2008b).
Figure 9 shows EC/V and OC/V versus V at LMP and CGR. Similarly to sulfate and
nitrate, OC/V decreases with increasing V and appears to reach a minimum value for V
> 15 $ng/m^3$. As discussed in section 3.1, other OC sources in addition to ships are
probably present at CGR even at high values of V. Thus, we assume that the OC/V
value obtained at Lampedusa for V>15 $ng/m^3$ is representative of cases dominated by
ship emissions, and this ratio is used to estimate the OC contribution due to ships at
both sites.
The pattern of the ratio EC/V versus V is less clear; in particular, several very low values
of EC/V appear also at small values of V. This result is unexpected because V and EC
are both markers of the primary ship aerosol, but the data here presented seem to
suggest that non negligible EC contributions from other sources were present, or that
different fractionating effects acted during the transport.
**3.4 Contribution of the ship aerosol to $PM_{10}$**



With all the limitations above described, by using the lower limits for the ratios
$(nssSO_4^{2-}/V)$, $(NO_3^-/V)$, and $(OC/V)$ representative for ship aerosol it is possible to
estimate the minimum contribution of $nssSO_4^{2-}$, $NO_3^-$ and OC emitted by ships to the
total budget of these component, and also to the total $PM_{10}$ mass. It has to be noticed
that the aerosol quantification obtained by this method is a rough estimate useful to
constrain the ship aerosol contribution. In addition, due to possibly different
meteorological conditions and photochemical activity, such values cannot be applied in
general as they can vary spatially and seasonally.
The minimum ratio of each specie with respect to V, the minimum estimated
contribution of ship emissions, for the average amount and for the maxima, to the total
concentration of these species and to $PM_{10}$, are reported in Table 3. As previously
discussed, the measured OC contribution is multiplied by 2.1 at LMP and by 1.8 at CGR
to obtain the total organic aerosol contribution.
At LMP, the estimated minimum concentration of non-sea-salt sulfate from ship
emissions was 1.3 µg/m$^3$, on average during this campaign. This value is lower than in
the previous study by Becagli et al. (2012) obtained over a longer period (2004-2008).
The relative contribution to the total sulfate is however similar here and in Becagli et al.
(2012), suggesting a similar role of $nssSO_4^{2-}$ from ship emissions to the total $nssSO_4^{2-}$
budget. At CGR the minimum ship contribution to sulfate, averaged over the same time
period, is higher than at  LMP (2.0 µg/m$^3$), but this higher value corresponds to a lower
contribution to the total $nssSO_4^{2-}$, confirming that other $nssSO_4^{2-}$ sources are important
at CGR.
Marmer and Langmann (2005) estimate that ship emissions contribute by 50% to the
total amount of $nssSO_4^{2-}$ in the Mediterranean. This value is larger than the one we
derive (about 30%); the reader is however reminded that we estimate a lower limit for
the ship contribution, useful to constrain the ship impact.
However, our data show that in cases with largest ship impact the $nssSO_4^{2-}$ from ship
contributes at least by 66% and 75% to the total $nssSO_4^{2-}$ at  LMP and CGR,
respectively.
Ships appear to contribute by small fractions to the total budget of $NO_3^-$. As previously
mentioned, the atmospheric chemistry of $NO_3^-$ is complex and the contribution of nitrate
from ship emission could be highly variable especially in the Mediterranean region
where high amount of UV radiation and highly reactive radical species are present.
Organic aerosol from ships also contributes significantly to the total OA amount and to
the total PM; in particular, at LMP at least about 92% of the total OA may be attributed
to the ship source in the case with maximum ship impact.
By summing these three contributions, it is possible to estimate the total aerosol mass
due to ship emissions, and its contribution to the total mass of $PM_{10}$. The lower limit for
the ship contribution was 1.9 µg/m$^3$ and 2.8 µg/m$^3$, corresponding to 11% and 8.2% of
$PM_{10}$ at LMP and CGR, respectively.



These percent contributions are higher than the annual average for the Mediterranean
Region estimated by Viana et al. (2014). It has to be considered that these authors
used data from harbour or coastal sites, which are highly affected by other sources in
addition to ships, and where gas-to-particle conversion is still at its initial phase.
Moreover, the percentage reported in this study refers to the summer season, when the
ship contribution in the Mediterranean region is higher (Becagli et al., 2012).
In cases with maximum ship impact, the estimated lower limit for the ship contribution
was between 40% and 48% of the total $PM_{10}$.

**Summary and conclusions**

In this study, we investigate the impact of the ship emissions to $PM_{10}$ on measurements
made at two sites in the central Mediterranean. The main objectives of the study were
to unambiguously identify the tracers of ship emissions in the sampled aerosol, and to
obtain a lower limit for the produced impact.
The $PM_{10}$ samples were collected in summer 2013, as a contribution to the Chemistry
and Aerosol Mediterranean Experiment, in parallel at Lampedusa and at Capo Granitola,
respectively South and North of the main shipping route through the Mediterranean.
The identification of aerosol originating from ships was based on an integrated analysis
combining chemical analyses, calculations of backward trajectories using a high
resolution regional model, and on tracking of ship traffic in the Mediterranean through
the Automatic Identification System.
The main results of this study may be summarized as follows:
1. moderate and elevated values of V and Ni in the aerosol were unambiguously
associated with the ship source; this attribution was based on:
- the V to Ni ratio, which corresponds to what expected for heavy fuel oil
771       combustion;

- low amounts of La and Ce with respect to V, and La/Ce ratio similar to those in
773       the UCC, which allowed to exclude power plants or refineries as sources
774       significantly contributing to the observed aerosol;

- coincidences between air mass trajectories and travelling ships;
2. in addition to travelling ships, also the planetary boundary layer vertical structure
played an important role in determining the dispersion of aerosols from the ship
source; temperature inversions appeared to be associated with elevated amounts of
ship emissions tracers, suggesting that they favoured the build-up of aerosol
concentration in the lowest atmospheric layers;




3. merchant ships (cargo and tankers) appeared to produce a larger impact on the measured aerosol than fishing vessels;

4. lower limits for the ratios $nssSO_4^{2-}/V$, $NO_3^-/V$, and OC/V, identifying the ship-dominated emission cases, were derived from the observations. The lower limits are respectively 200, 10, and 40. These lower limits are expected to be season and site-dependent;

5. by using these ratios, the lower limits to the contribution of the ship source to $nssSO_4^{2-}$, $NO_3^-$, OA, and to $PM_{10}$ during the measurement campaign were estimated. Ship emissions contributed by at least 30% to the total amount of sulfate, by at least 4-7% to the total amount of $NO_3^-$, and by at least 8-14% to the total amount of organic aerosol. All these contributions correspond at least to 11% of $PM_{10}$ at LMP (1.9 $\mu g/m^3$), and about 8% of $PM_{10}$ at CGR (2.8 $\mu g/m^3$). In cases with largest ship impact, ships contributed up to 12 $\mu g/m^3$ to $PM_{10}$, and by about 48% of $PM_{10}$ at LMP and 40% at CGR.

## Acknowledgements

Measurements at Lampedusa were partly supported by the Italian Ministry for University and Research through the NextData and Ritmare projects.
We thank the Institute for Coastal Marine Environment of the National Research Council (IAMC-CNR), for hosting the instruments at Capo Granitola. Thanks are due to MarineTraffic (www.marinetraffic.com) for providing the information on the ship traffic in the Sicily Channel.



**Table 1**. Mean $PM_{10}$ load and composition with the related standard deviation and
percentage with respect to $PM_{10}$ (in bracket) at Lampedusa and Capo Granitola. Mean,
standard deviation and percentage are calculated on homogeneous data sets for both
sites considering all the common sampling ("all data" columns) and excluding the
mistral events ( "Mistral excluded" columns).

| | Lampedusa | | Capo Granitola | |
|---|---|---|---|---|
| | **All data** | **Mistral excluded** | **All data** | **Mistral excluded** |
| **$PM_{10}$ ($\mu g/m^3$)** | 18.0±6.6 | 16.3±5.2 | 34.1±18.9 | 27.2±6.5 |
| **Sea Salt Aerosol ($\mu g/m^3$)** | 4.63±6.30 (25.7%) | 2.33±3.21 (14.3%) | 8.14±15.50 (23.9%) | 2.12±6.51 (7.8%) |
| **Crustal Aerosol ($\mu g/m^3$)** | 0.82±0.44 (4.6%) | 0.90±0.43 (5.5%) | 2.80±1.7 (8.2%) | 3.02±1.75 (11.1%) |
| **$nssSO_4^{2-}$ ($\mu g/m^3$)** | 3.95±2.28 (21.9%) | 4.40±2.22 (27.0%) | 6.78±3.08 (19.9%) | 7.53±2.78 (27.7%) |
| **$NH_4^+$ ($\mu g/m^3$)** | 0.98±0.56 (5.5%) | 1.09±0.55 (6.7%) | 1.48±0.94 (4.3%) | 1.66±0.87 (6.1%) |
| **$NO_3^-$ ($\mu g/m^3$)** | 1.25±1.00 (7.0%) | 1.02±0.02 (6.2%) | 1.35±1.11 (4.0%) | 1.01±0.82 (3.7%) |
| **OA ($\mu g/m^3$)** | 3.86±1.56 (21.4%) | 4.04±1.59 (24.8%) | 9.02±2.52 (26.5%) | 9.53±2.29 (35.0%) |
| **EC ($\mu g/m^3$)** | 0.15±0.08 (0.8%) | 0.15±0.08 (0.9%) | 0.44±0.28 (1.3%) | 0.51±0.26 (1.9%) |
| **Unknown ($\mu g/m^3$)** | 2.52±3.26 (14.0%) | 2.20±3.40 (13.5%) | 4.11±7.78 (12.1%) | 1.82±4.48 (6.7%) |




**Table 2.** Correlation parameters between V and Ni at LMP and CGR $PM_{10}$ samples for
all the samples and for samples with V concentration higher than 6 ng/m$^3$.

|  |  | Slope (± uncertainty) | $R^2$ | n. |
|---|---|---|---|---|
| **LMP** | All data | 2.94±0.03 | 0.986 | 124 |
|  | $V_{sol}$ > 6ng/m$^3$ | 2.99±0.03 | 0.994 | 44 |
| **CGR** | All data | 2.82±0.08 | 0.950 | 59 |
|  | $V_{sol}$ > 6ng/m$^3$ | 3.00±0.05 | 0.989 | 34 |

**Table 3.** Estimated minimum ratio of $nssSO_4^{2-}$, $NO_3^-$, OA with respect to V, minimum
contribution of each species from ship emissions averaged over the considered time
period and for the cases with highest ship impact of $nssSO_4^{2-}$, $NO_3^-$, OA and $PM_{10}$ at
LMP and CGR.

| | $nssSO_4^{2-}$ | | $NO_3^-$ | | OA | | $PM_{10}$ | |
|---|---|---|---|---|---|---|---|---|
| | $(nssSO_4^{2-}/V)_{min}$=200 | | $(NO_3^-/V)_{min}$=10 | | $(OC/V)_{min}$=40 | | | |
| | LMP | CGR | LMP | CGR | LMP | CGR | LMP | CGR |
| **Average contribution µg/m$^3$ (%)** | 1.3 (33%) | 2.0 (30%) | 0.065 (4.5%) | 0.10 (7.2%) | 0.55 (14%) | 0.72 (8.1%) | 1.9 (11%) | 2.8 (8.2%) |
| **Maximum contribution µg/m$^3$ (%)** | 7.2 (66%) | 8.6 (75%) | 0.36 (50%) | 0.43 (80%) | 3.0 (92%) | 3.1 (21%) | 10.6 (48%) | 12.1 (40%) |

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
