# Peer review of "Constraining the ship contribution to the aerosol of the Central Mediterranean"

_Atmospheric Chemistry and Physics, 2016_

## Referee Comment (RC1) · Anonymous Referee #1 · 26 Jul 2016

Dear Editor,

this manuscript presents an assessment of shipping contributions to PM10 aerosols in two locations in the Mediterranean Sea. It presents a very interesting integrated approach combining different tools such as analysis of the chemical composition of PM10, tracer analysis, back-trajectories and ship inventories and databases. Whereas certain of these tools are not novel and suffer from limitations (e.g., tracer methods), the combination of all of them provides very interesting results. Over all, the paper is of interest to the scientific community and merits publication after revision. Specific comments are provided below:

- lines 110-113 would fit better in the Methods section

- line 276: very good, it is very important that the authors highlight this kind of limitation

[Figure]

- line 292: "smaller" should be "lower". In general, the English is correct but a review by a native speaker would be helpful.

- Figure 2: the nitrate contribution at LMP is surprisingly high, considering the thermal instability of this species. Can this be due to an artefact or high uncertainty of the measurements?

- line 340: also, EC concentrations are much higher in CGR, supporting this interpretation (the dominance of anthropogenic sources of carbonaceous aerosols in CGR)

- Figure 4 is interesting but not so useful, it could be removed if the authors encounter space limitations

- line 402: about the representativeness of this kind of ratios: it should be stated that this ratio may have a large variability due to varying fuel composition and engine operating conditions

- line 424: LCR=1-5, how specific are these values for the specific refinery in the Moreno et al study? Can they really be extrapolated to other refineries? What can the uncertainty be for other refineries? Please discuss

- line 430: 24% of samples with LCR>1 and 8% >2, this is a very high contribution (32%), can the contribution from refineries be so high at LMP? From where? Or is this an example of the limitation mentioned above, the uncertainty of these LCR values when applied to other study areas? Please discuss

- Figure 6: again, the authors show only one point per type of dust, what is the uncertainty? The authors' results seem very conclusive with regard to their own samples, but it would still be useful to see the potential variability in the other (refinery, UCC) types of dust. If no other data is available for this kind of sources, please mention as a limitation

- line 575: it is also possible that this is due to the larger aerosol dilution during transport towards LMP?

- line 666: "two times higher than...", isn't this strongly dependent on fuel and engine operating conditions?

- line 702: "component" should be "components"

- line 703: the term "rough estimate" is more adequate than the "unambiguous identification" used in other parts of the text

- line 722: "50%", again, this is very variable depending on engine conditions, meteorology (oxidation rate of SO2...), etc. Therefore this 20% difference can be expected

- line 741: the higher % contributions obtained should not be considered a negative result, due to the potentially large variability of these contributions. It is correct to compare the order of magnitude with the previous literature, but the precision of the authors' method (and of other currently available methods) is not sufficient for them to carry out a detailed comparison of the results obtained. The results presented are quite positive, in my opinion.

- line 781: should "as expected", be added at the start of the sentence?

---

## Referee Comment (RC2) · Anonymous Referee #2 · 26 Aug 2016

This is a nice work that attempts to provide some estimates of the contribution of ship emissions to the budget of specific chemical species as well as to PM10 in total. There is undoubtedly quite a large fraction of uncertainty into this, as the authors themselves admit and partly discuss, which is based not only on the methodological limitations but also on the short period of this study and the very specific spatial grid they are referring at. In this line, even though I recognize some weakness of the paper to provide substantial new concepts, ideas or methods (as required by ACP), I still see the importance of such local studies in a hot issue like ship emissions in the Mediterranean and support its appropriateness for publication in the CHARMEX SI. I am suggesting major revision mainly because I would like to urge the authors to reorganize the presentation of their results in a way that they show up better the important aspects coming out of it.

General comment: As the authors mention in lines 704-706, the use of their results

(they are referring to the V different ratios but this is transferred also to the rest of the analysis) is limited by the fact that they refer to specific meteorology, photochemistry, space and season. This is obvious and unavoidable, but still there should be some discrimination of what is indeed only of local interest and what could be somewhat generalized e.g. by using ranges of values, relations with the level of traffic, or better describing the sensitivity of those results and comparing them more detailed with existing literature (even their own previous work). Moreover, the structure of the paper, at some point, confused me by means that I was expecting a number of synergistic analyses to identify and discriminate ship emissions (e.g. the standard markers, then some improvement by rare earth elements, then further refinement via trajectories etc), while Figure 8 speaks for itself! The authors might like to start from Fig. 8 and then provide further poof or try to explain discrepancies e.g. between the two sites. My overall impression is that the paper started from V and after a loop of some necessary and some unnecessary steps it ends up again with V. I hope some of my more specific comments that follow will help authors make improvements to the manuscript.

Specific comments:

1. The language is very clear to understand but it should definitely benefit from some native English speaker editing.

2. The word "anthropic" is probably misused instead of "anthropogenic". This is not based on the frequency of use of the words in the relevant literature, but mainly on the meaning of "anthropic" which is more what is influenced by humans or taking place during human era, rather than what is generated by human activities ... I see its use mainly in social sciences. In Greek they use the translation of the word anthropogenic and not anthropic to refer to sources or pollution (in any case they are both Greek words!).

3. Part of the paragraph in lines 100-118 seem to fit better as introduction to section 2.1. That would also include Fig.1 and the description of the sites. I would suggest to

keep in the introduction only the nice unfolding of the diachronically followed strategy.

4. In section 2.1 (lines 146-153) there is a great number of references to available measurement types at Lambedusa that seem irrelevant to the specific work. Only references contributing to the description of Lambedusa characteristics as a site should be included here.

5. In section 2.1 there is a lot of information about the techniques used in the two stations. Probably the authors would like to organize part of this info in a table, to help readers follow easier.

6. Please add information on start-end times of the 12h sampling at the two sites. See e.g. line 172, what is meant by diurnal sampling?

7. Section 2.3 (lines 228-232). Please check on Solomos et al. 2015 who are articulating improvements in resolving and forecasting the dispersion of smoke plumes, in their case, over particularly complex terrains (including BL and sea-breeze system particularities), by incorporating high-resolution (spatial and temporal) meteorology and satellite data. (Solomos S., V. Amiridis, P. Zanis, E. Gerasopoulos, F.I. Sofiou, T. Herekakis, J. Brioude, A. Stohl, R.A. Kahn, C. Kontoes, Smoke dispersion modeling over complex terrain using high resolution meteorological data and satellite observations – The FireHub platform, Atmospheric Environment, Volume 119, http://dx.doi.org/10.1016/j.atmosenv.2015.08.066.)

8. Section 3.1 (lines 276-278). Could you here quantify this low crustal elements contribution as a percentage? (see lines 295-296).

9. Section 3.1 (lines 283-284). A reference should be provided.

10. Section 3.1 (lines 286-288). Though I understand the meaning, this sentence should be rephrased.

11. Section 3.1 (lines 335-336, Fig. 3). I do not see this different behavior. At the same part of the plot which corresponds to lower values (see LMP axes) the behavior

seems similar – no specific correlation. So it is from a threshold and upwards that the influence of primary sources is shown on CGR. Please rearrange the interpretation.

12. Section 3.1 (line 356, Fig. 4). On June 18 the diurnal cycle is opposite to what is described. (lines 359-362) This is not the case in all days ... could the authors identify the sea breeze influenced days on the plot?

13. Section 3.1.2 (lines 429-432). The percentage for LCR>1 at LMP seems large. How much is it biased by the fact that low La and Ce (especially Ce) values could produce arbitrary ratios? Please put a threshold to refine this analysis. Is this analysis with LCR used somehow to further constrain your statistical sample in the following parts? Or is it just to prove the appropriateness of V as a marker in this period? If so, what is the improvement brought in and how does it compare with the trajectories analysis?

14. Section 3.2.1 (Fig. 7). The scale used in Figure 7 is probably not appropriate. I would avoid showing contributions lower than 1%, they take up most of the space in the plot without having anything to say about this. Choose the scale and probably the area to show, so that it fits to the local aspects that this part of the analyses addresses. How confident are you for the representation of the backtrajectories in the very short distances (few kilometers) and the such low altitudes (few hundred meters) between the trajectory starting points and the ship tracks? Once more, as in the previous comment, how it compares if e.g. you change the selection criteria, does it much the LCR results?.

15. Section 3.3 (Fig. 9). As far as I understood the authors have used a previous obtained value of 200 for nssSO4/V, confirmed by this shorter study, and then calculated similar ratios for other species only from this study. It seems to me more like an eye approach which does not take into account the uncertainty of the points and the log scale used. Thus, the value of 10 NO3/V could easily be 20 or 30. I would suggest a more thorough analysis on this, based on some logarithmic fitting and then extraction of a

plateau value for higher V. Then give ranges based on the uncertainty and propagate the error into the final calculations that follow.

16. Section 3.4. I think this is the part that my general comment should be mostly taken into account. Highlight what is of general interest than only of local impact and use; elaborate further on comparisons (e.g. like in lines 741-746).

---

## Author Comment (AC1) · 22 Oct 2016

Anonymous Reviewer #1

Dear Editor, this manuscript presents an assessment of shipping contributions to PM10 aerosols in two locations in the Mediterranean Sea. It presents a very interesting integrated approach combining different tools such as analysis of the chemical composition of PM10, tracer analysis, back-trajectories and ship inventories and databases. Whereas certain of these tools are not novel and suffer from limitations (e.g., tracer methods), the combination of all of them provides very interesting results. Over all, the paper is of interest to the scientific community and merits publication after revision.

Specific comments are provided below: - lines 110-113 would fit better in the Methods section

[Figure]

Figure 1 and the corresponding text were moved to the methods section as suggested.

- line 276: very good, it is very important that the authors highlight this kind of limitation.

- line 292: "smaller" should be "lower". In general, the English is correct but a review by a native speaker would be helpful.

The sentence was corrected. The paper was revised by a native English speaker.

- Figure 2: the nitrate contribution at LMP is surprisingly high, considering the thermal instability of this species. Can this be due to an artefact or high uncertainty of the measurements? The nitrate concentrations measured in this campaign are in agreement with the long term measurements performed at Lampedusa (e.g., Calzolai et al., 2015) and with data from other remote sites in the western (Mallorca; e.g. Simo et al., 1991) and eastern Mediterranean (Finokalia; e.g. Mihalopoulos et al., 1997). The presence of artefacts in the nitrate sampling are documented in literature. These artefacts cause lower concentrations in the sample than in the atmosphere, in polluted regions due to NH4NO3 volatilization deposited on the filter. We believe than in relatively clean conditions like those incurring in Lampedusa and Capo Granitola the impact of artefacts is small, and the derived values are reliable. These values also suggest that ship emissions may play a large role. A comment was added to the text.

- line 340: also, EC concentrations are much higher in CGR, supporting this interpretation (the dominance of anthropogenic sources of carbonaceous aerosols in CGR)

We agree with the reviewer (see also comment 11 of reviewer 2). The sentence was changed and the differences were discussed..

- Figure 4 is interesting but not so useful, it could be removed if the authors encounter space limitations

We agree with the reviewer; the discussion of the sea breeze is interesting, but not directly related with the paper main topic. We moved the figure to the supplementary material.

- line 402: about the representativeness of this kind of ratios: it should be stated that this ratio may have a large variability due to varying fuel composition and engine operating conditions

We agree with the reviewer. The sentence was modified.

- line 424: LCR=1-5, how specific are these values for the specific refinery in the Moreno et al study? Can they really be extrapolated to other refineries? What can the uncertainty be for other refineries? Please discuss

Section 3.1.2 was largely modified, also taking into account the different comments of the reviewers. The discussion on the LCR variability was improved and new references to support the discussion were added. In particular, we have reported range of values for ship, refinery, and crustal components of the aerosols, also with reference to literature results.

- line 430: 24% of samples with LCR>1 and 8% >2, this is a very high contribution (32%), can the contribution from refineries be so high at LMP? From where? Or is this an example of the limitation mentioned above, the uncertainty of these LCR values when applied to other study areas? Please discuss

We have double checked the calculated LCR values. As correctly suggested by reviewer 2 (see his/her point 13), we did not take into account the uncertainty and the detection limit of La and Ce measurements. The instrumental detection limit on these determination was added in section 2.1. The very high values of LCR correspond with very low, in some cases below the detection limit, La and/or Ce concentrations. In these cases the uncertainty on LCR exceeds 100%. The elimination of these data points filters out most of the very high LCR values. We have thus redrawn Figure 5 (that becomes fig 4) accordingly, also indicating the range of values which are expected for the crustal component. This figure is reported below.

- Figure 6: again, the authors show only one point per type of dust, what is the uncertainty? The authors' results seem very conclusive with regard to their own samples, but it would still be useful to see the potential variability in the other (refinery, UCC) types of dust. If no other data is available for this kind of sources, please mention as a limitation

The Lanthanoid content in Saharan dust measured in several aerosol size classes and from different areas of Sahara, as well for other compounds are reported in figure 6 (figure 5 in the revised version). The discussion on the variability of the composition was expanded (see also answer to the two previous comments). The figure is reported below.

- line 575: it is also possible that this is due to the larger aerosol dilution during transport towards LMP?

We agree with the reviewer. Dilution may be a cause of lower V concentration at LMP. A sentence on the possible role of dilution was added

- line 666: "two times higher than...", isn't this strongly dependent on fuel and engine operating conditions?

We agree. The sentence was changed accordingly.

- line 702: "component" should be "components"

The typo was corrected.

- line 703: the term "rough estimate" is more adequate than the "unambiguous identification" used in other parts of the text

In the text we use "rough estimation" referring to the quantitative estimation of the minimum contribution of ship aerosol (i.e., using the method of the minimum values reported in section 3.4). However, we believe that the combination of the different methods discussed in the paper leads to an unambiguous attribution of the measured aerosol to the ship source.

- line 722: "50%", again, this is very variable depending on engine conditions, meteorology (oxidation rate of SO2...), etc. Therefore this 20% difference can be expected

We agree. This is one of the reasons why we may estimate only the lower limit for sulfate contribution. The sentence was corrected.

- line 741: the higher % contributions obtained should not be considered a negative result, due to the potentially large variability of these contributions. It is correct to compare the order of magnitude with the previous literature, but the precision of the authors' method (and of other currently available methods) is not sufficient for them to carry out a detailed comparison of the results obtained. The results presented are quite positive, in my opinion.

We fully agree. The sentence is explaining possible reasons for the observed differences. - line 781: should "as expected", be added at the start of the sentence? We agree with the reviewer's comment, and added "as expected" at the beginning of the sentence.
* * *
[Figure]

**Fig. 1.** Figure 4

Legend:
- ○ CGR
- ○ LMP
- ■ UCC - Henderson and Henderson, 2009
- ▲ Refinery - Olmez and Gordon, 1985
- ◆ Sahara - Castillo et al., 2008
- ◆ Sahara - Moreno et al., 2006
- ▶ FCC - Kulkarni et al., 2006
- ● Moreno et al 2008_PM10
- ● Moreno et al 2008_PM2.5
- ○ CGR-crust
- ○ CGR La/Ce>1
- ○ LMP La/Ce>1

Axes: La*3.1, Ce*1.54, V

**Fig. 2.** figure 5

---

## Author Comment (AC2) · 22 Oct 2016

Anonymous Reviewer #2 This is a nice work that attempts to provide some estimates of the contribution of ship emissions to the budget of specific chemical species as well as to PM10 in total. There is undoubtedly quite a large fraction of uncertainty into this, as the authors themselves admit and partly discuss, which is based not only on the methodological limitations but also on the short period of this study and the very specific spatial grid they are referring at. In this line, even though I recognize some weakness of the paper to provide substantial new concepts, ideas or methods (as required by ACP), I still see the importance of such local studies in a hot issue like ship emissions in the Mediterranean and support its appropriateness for publication in the CHARMEX SI. I am suggesting major revision mainly because I would like to urge the authors to reorganize the presentation of their results in a way that they show up better

the important aspects coming out of it.

General comment: As the authors mention in lines 704-706, the use of their results(they are referring to the V different ratios but this is transferred also to the rest of the analysis) is limited by the fact that they refer to specific meteorology, photochemistry, space and season. This is obvious and unavoidable, but still there should be some discrimination of what is indeed only of local interest and what could be somewhat generalized e.g. by using ranges of values, relations with the level of traffic, or better describing the sensitivity of those results and comparing them more detailed with existing literature (even their own previous work).

We only partly agree with the comments on the local nature of the study. Of course, the analysis refers to a specific region, meteorology, season, etc. However, we believe that the information we retrieve is sufficiently robust, and relevant on a larger scale than local, mainly because it is based on two sites with markedly different characteristics. In particular, the site of Lampedusa is relatively far from the main shipping route, and the observations there suggest that ship emissions impact the Mediterranean basin on a larger scale than local. We have however added some comments and references with the aim of placing the obtained results in a wider perspective.

Moreover, the structure of the paper, at some point, confused me by means that I was expecting a number of synergistic analyses to identify and discriminate ship emissions (e.g. the standard markers, then some improvement by rare earth elements, then further refinement via trajectories etc), while Figure 8 speaks for itself! The authors might like to start from Fig. 8 and then provide further poof or try to explain discrepancies e.g. between the two sites. My overall impression is that the paper started from V and after a loop of some necessary and some unnecessary steps it ends up again with V. I hope some of my more specific comments that follow will help authors make improvements to the manuscript.

We agree that links among the different parts of the analysis are lacking or have been

poorly discussed. However, we believe that including these lacking links the logical structure of the paper is sound, and we did not change the overall structure of the paper in the revised version. We have explained in more detail the connections among the different analyses, and the way in which they affect data. Some parts were heavily reorganized (see e.g., section 3.1.2), and some non essential information was removed (e.g., figures 4). We believe that the presentation is now clearer, and thank the reviewer for this suggestion.

Specific comments: 1. The language is very clear to understand but it should definitely benefit from some native English speaker editing.

The paper was revised by a native English speaker. . 2. The word "anthropic" is probably misused instead of "anthropogenic". This is not based on the frequency of use of the words in the relevant literature, but mainly on the meaning of "anthropic" which is more what is influenced by humans or taking place during human era, rather than what is generated by human activities ... I see its use mainly in social sciences. In Greek they use the translation of the word anthropogenic and not anthropic to refer to sources or pollution (in any case they are both Greek words!).

We agree with the comment. We have checked te use of the terms "anthropic" and "anthropogenic" throughout the paper.

3. Part of the paragraph in lines 100-118 seem to fit better as introduction to section 2.1. That would also include Fig.1 and the description of the sites. I would suggest to keep in the introduction only the nice unfolding of the diachronically followed strategy.

Figure 1 and the text were moved to section 2.

4. In section 2.1 (lines 146-153) there is a great number of references to available measurement types at Lambedusa that seem irrelevant to the specific work. Only references contributing to the description of Lambedusa characteristics as a site should be included here.

The number of references was reduced as suggested.

5. In section 2.1 there is a lot of information about the techniques used in the two stations. Probably the authors would like to organize part of this info in a table, to help readers follow easier.

A table with the sampling strategy and measurements carried out on each filter was added.

6. Please add information on start-end times of the 12h sampling at the two sites. See e.g. line 172, what is meant by diurnal sampling?

The start and end sampling time was added in the text and in the table reporting the sampling strategy.

7. Section 2.3 (lines 228-232). Please check on Solomos et al. 2015 who are articulating improvements in resolving and forecasting the dispersion of smoke plumes, in their case, over particularly complex terrains (including BL and sea-breeze system particularities), by incorporating high-resolution (spatial and temporal) meteorology and satellite data. (Solomos S., V. Amiridis, P. Zanis, E. Gerasopoulos, F.I. Sofiou, T. Herekakis, J. Brioude, A. Stohl, R.A. Kahn, C. Kontoes, Smoke dispersion modeling over complex terrain using high resolution meteorological data and satellite observations – The FireHub platform, Atmospheric Environment, Volume 119, http://dx.doi.org/10.1016/j.atmosenv.2015.08.066.)

A comment and the reference were added in the discussion.

8. Section 3.1 (lines 276-278). Could you here quantify this low crustal elements contribution as a percentage? (see lines 295-296).

The percentage of crustal content with respect to the total PM10 concentration was added.

9. Section 3.1 (lines 283-284). A reference should be provided.

[Figure]

The sentence applies to Lampedusa (or remote marine sites). This is now specified in the text.

10. Section 3.1 (lines 286-288). Though I understand the meaning, this sentence should be rephrased.

We agree. The sentence was rearranged..

11. Section 3.1 (lines 335-336, Fig. 3). I do not see this different behavior. At the same part of the plot which corresponds to lower values (see LMP axes) the behaviour seems similar – no specific correlation. So it is from a threshold and upwards that the influence of primary sources is shown on CGR. Please rearrange the interpretation.

We agree with the reviewer. Essentially, there are different concentration ranges at the two sites, and a correlation for moderate and elevated values at CGR. The sentence was corrected. We have changed the sentence.

12. Section 3.1 (line 356, Fig. 4). On June 18 the diurnal cycle is opposite to what is described. (lines 359-362) This is not the case in all days ... could the authors identify the sea breeze influenced days on the plot?

The discussion of the evolution linked to sea breeze and coastal circulation, and the different daily evolutions go beyond the main objective of the paper. Figure 4 was consequently moved to the supplementary material..

13. Section 3.1.2 (lines 429-432). The percentage for LCR>1 at LMP seems large. How much is it biased by the fact that low La and Ce (especially Ce) values could produce arbitrary ratios? Please put a threshold to refine this analysis. Is this analysis with LCR used somehow to further constrain your statistical sample in the following parts? Or is it just to prove the appropriateness of V as a marker in this period? If so, what is the improvement brought in and how does it compare with the trajectories analysis?

We agree. The very high LCR values are discussed in the answer to reviewer 1 (see

comment to line 430). The impact of the LCR values on the selected data was added, as well a discussion with respect to the trajectory analysis. The impact of LCR is shown in the following figures and in the dataset, and is discussed in the following parts of the paper. The cases with elevated values of LCR and possibly influenced by refineries have been highlighted also in figure 8 (figure 7 in the revised manuscript).

14. Section 3.2.1 (Fig. 7). The scale used in Figure 7 is probably not appropriate. I would avoid showing contributions lower than 1%, they take up most of the space in the plot without having anything to say about this. Choose the scale and probably the area to show, so that it fits to the local aspects that this part of the analyses addresses. How confident are you for the representation of the backtrajectories in the very short distances (few kilometers) and the such low altitudes (few hundred meters) between the trajectory starting points and the ship tracks? Once more, as in the previous comment, how it compares if e.g. you change the selection criteria, does it much the LCR results?.

We understand the reviewer's point but the purpose of Figure 7 was just to give an overview of the results of the trajectory analysis and to show the air masses origin on a continental scale rather than a local one, as a consequence of the prevailing weather regimes observed during the campaign (e.g. the relatively high frequency of air masses coming from Central and Western Europe, especially in June 2013 due to the numerous Mistral episodes). In fact, Section 3.2.1 serves as an introduction to the more detailed analysis focused on the local aspects, which is discussed in the following Section 3.2.2. Furthermore, we cannot fully agree that contributions lower than 1% take up most of the scale in the plot, therefore we would rather keep Figure 7 (now figure 6) unchanged. As the Reviewer correctly points out, the calculation of backtrajectories in the lowest layers and at short distances from the starting point may be affected by substantial uncertainties. However, this is particularly true when using coarse meteorological fields from global models (resolution of order of 50 km in space and three or six hours in time). In the present study outputs from a mesoscale model with relatively

high spatial resolution (10 km and 35 vertical levels) and hourly temporal resolution are used to drive trajectory calculations. Though for some more specific applications and in particularly complex topography areas even higher resolutions would be required, we believe that for the purposes of our study the quality of meteorological fields is more than adequate. Indeed, our main focus is the description of transport within the marine boundary layer, which is reasonably well described using a 10-km grid spacing.

15. Section 3.3 (Fig. 9). As far as I understood the authors have used a previous obtained value of 200 for nssSO4/V, confirmed by this shorter study, and then calculated similar ratios for other species only from this study. It seems to me more like an eye approach which does not take into account the uncertainty of the points and the log scale used. Thus, the value of 10 NO3/V could easily be 20 or 30. I would suggest a more thorough analysis on this, based on some logarithmic fitting and then extraction of a plateau value for higher V. Then give ranges based on the uncertainty and propagate the error into the final calculations that follow.

We agree with the reviewer that the limit ratio was determined arbitrarily. In order to obtain a better estimate, we have tried to follow the reviewer's suggestion and have tried to fit different analytical curves to the data. However, we could not find a coherent fitting scheme for the different species, and ended up in choosing a simpler method. The limit ratio was calculated as the average of the observed values for V > 15 ng/m3. This method allows to derive a quantitative estimate of the limit ratio, and to calculate its standard deviation. Since we are determining a lower limit for the ship contribution to the total aerosol load, and a values of the ratio may still be decreasing for V around 15 ng/m3, we used for each species a limit value which is equal to the average minus one standard deviation. These limit values are somewhat different at Lampedusa and Capo Granitola, and are reported in figure 9 (now figure 8) and discussed in the text.

16. Section 3.4. I think this is the part that my general comment should be mostly taken into account. Highlight what is of general interest than only of local impact and use; elaborate further on comparisons (e.g. like in lines 741-746).

We have expanded the discussion at the end of section 3.4. See also the answer to reviewer 2 general comment.

[Figure]

[Figure]

**Fig. 1.** Figure 4

[Figure]

Legend:
- CGR
- LMP
- UCC - Henderson and Henderson, 2009
- Refinery - Olmez and Gordon, 1985
- Sahara - Castillo et al., 2008
- Sahara - Moreno et al., 2006
- FCC - Kulkarni et al., 2006
- Moreno et al 2008_PM10
- Moreno et al 2008_PM2.5
- CGR-crust
- CGR La/Ce>1
- LMP La/Ce>1

**Fig. 2.** figure 5